# DIFFERENTIABLE BAYESIAN NEURAL NETWORK INFERENCE FOR DATA STREAMS

## ABSTRACT

While deep neural networks (NNs) do not provide the confidence of its prediction, Bayesian neural network (BNN) can estimate the uncertainty of the prediction. However, BNNs have not been widely used in practice due to the computational cost of predictive inference. This prohibitive computational cost is a hindrance especially when processing stream data with low-latency. To address this problem, we propose a novel model which approximate BNNs for data streams. Instead of generating separate prediction for each data sample independently, this model estimates the increments of prediction for a new data sample from the previous predictions. The computational cost of this model is almost the same as that of non-Bayesian deep NNs. Experiments including semantic segmentation on real-world data show that this model performs significantly faster than BNNs, estimating uncertainty comparable to the results of BNNs.

## 1 INTRODUCTION

While deterministic neural networks (DNNs) surpass human capability in some area in terms of prediction accuracy (He et al., 2015; Silver et al., 2016; Ardila et al., 2019), it could not estimate a trustworthy uncertainty of prediction. Since the prediction can not be perfect and the misprediction might result in fatal consequences in areas such as medical analysis and autonomous vehicles control, estimating uncertainty as well as predictions will be crucial for the safer application of machine learning based systems.

Bayesian neural network (BNN), a neural network (NN) that uses probability distributions as weights, estimates not only predictive results but also uncertainties. This allows computer systems to make better decisions by combining uncertainty with prediction. Moreover, BNN can achieve high performance in a variety of fields, e.g. image recognition (Kendall et al., 2015; Kendall & Gal, 2017), language modeling (Fortunato et al., 2017), reinforcement learning (Kahn et al., 2017; Osband et al., 2018), meta-learning (Yoon et al., 2018; Finn et al., 2018), and multi-task learning (Kendall et al., 2018), by exploiting uncertainty.

Although BNNs have these theoretical advantages, they have not been used as a practical tool. The predictive inference speed has detained BNNs from wide applications. BNN executes NN inference for dozens of samples from weight distributions. Since sampling and multiple NN executions are difficult to be parallelized, the inference execution takes an order of magnitude more time. Particularly, this is a significant barrier for processing data streams with low-latency.

Most time-varying data streams change continuously, and so do the predictions of BNNs. Thus, we estimate prediction by calculating the increments between two consecutive results, instead of calculating the separate prediction for each input data. This is equivalent to calculating the differentiation of the BNN's prediction for arbitrary data, because the difference of prediction for the new data is the line integration of the gradient of prediction over the new data.

In this work, we propose a *differentiable BNN* (DBNN) inference with respect to an input data. The prediction of the DBNN is given by a Monte Carlo (MC) estimator for distributions of data streams and weights. We speed up the inference by approximating the distribution using histogram and calculating the gradient for this MC estimator. We show that the time complexity of this model is nearly the same as that of deep DNNs. We evaluate DBNN with semantic segmentation using road scene video sequences and the results show that DBNN has almost no degradation in computational

performance compared to deep DNNs. The uncertainty predicted by DBNN is comparable to that of BNN in various situations.

The main contributions of this work are as follows.

- We propose online codevector histogram that estimates the probability of a high-dimensional data stream. Then, we show that this histogram can be used to obtain the MC gradient estimation.
- We propose differentiable Bayesian neural network inference with respect to input data as an approximation of Bayesian neural network inference for data streams. DBNN uses the online codevector histogram to approximate probabilities. This model is nonparametric and applied to trained BNN without significant modifications.
- We theoretically and empirically show that the computational performance of DBNN is almost the same as that of deep DNNs.

## 2 BACKGROUND AND RELATED WORK

BNNs are a state-of-the-art method to estimate predictive uncertainty while DNNs based approaches have been developed recently (Lakshminarayanan et al., 2017; Guo et al., 2017). BNNs with probability distributions as weights produce probabilistic results. To make prediction, BNNs sample from the weight probabilities and performs DNN for each sample. In this section, we describe the details and challenges of BNNs inference process.

### 2.1 BAYESIAN NEURAL NETWORK INFERENCE

Suppose that $p(\boldsymbol{w})$ is posterior probability of NN weights $\boldsymbol{w}$ and $p(\boldsymbol{y}|\boldsymbol{x}, \boldsymbol{w})$ is the BNN model for input data $\boldsymbol{x}$. Then, the inference result of BNN is a predictive distribution:

$$p(\boldsymbol{y}|\boldsymbol{x}_\star) = \int p(\boldsymbol{y}|\boldsymbol{x}_\star, \boldsymbol{w})p(\boldsymbol{w})d\boldsymbol{w} \tag{1}$$

where $\boldsymbol{x}_\star$ is observed input data vector and $\boldsymbol{y}$ is output vector. For simplicity, BNNs are usually modeled to have a probability distribution with the mean of the prediction of DNN, e.g. in Hernández-Lobato & Adams (2015); Gal & Ghahramani (2016):

$$p(\boldsymbol{y}|\boldsymbol{x}_\star, \boldsymbol{w}) = \mathcal{N}(\boldsymbol{y}|\text{NN}(\boldsymbol{x}_\star, \boldsymbol{w}), \tau^{-1}) \tag{2}$$

where $\text{NN}(\cdot)$ is prediction of DNN and $\tau$ is a given parameter. (1) can be approximated using MC estimator:

$$p(\boldsymbol{y}|\boldsymbol{x}_\star) \simeq \frac{1}{N_{\boldsymbol{w}_\star}} \sum_{\boldsymbol{w}_\star} p(\boldsymbol{y}|\boldsymbol{x}_\star, \boldsymbol{w}_\star) \tag{3}$$

where $\boldsymbol{w}_\star \sim p(\boldsymbol{w})$ and $N_{\boldsymbol{w}_\star}$ is the number of samples. The expected value of the obtained predictive distribution is the predictive result of BNN and the variance is the predictive uncertainty. The process of calculating the equation consists of two steps: sampling weights from $p(\boldsymbol{w})$, e.g. using Markov chain Monte Carlo (MCMC) (Neal et al., 2011; Hoffman & Gelman, 2014), and executing the DNN for each weight sample. Since real-world data is large and practical NNs are deep, the repetitive computation of NNs is difficult to be fully parallelized due to various problems, such as memory limitations of the GPU (Kendall & Gal, 2017), and the iterative computation of DNNs results in the decrease of computation speed. MCMCs are slow and, despite recent achievements (Tran et al., 2018), it is challenging to achieve linearly scaling performance for multi-GPU MCMC. In addition, it is difficult to check for the convergence of MCMCs. To mitigate these problems, we approximate the difference in the prediction for one new data as one NN calculation, and sample the weights from the posterior before testing.

Several sampling-free BNNs, e.g. Hernández-Lobato & Adams (2015); Wang et al. (2016); Hwang et al. (2018); Wu et al. (2018), have been proposed recently. The sampling-free BNNs predict results with only one forward pass. These methods, however, have limitations. First, sampling-free BNNs necessitate specific types of weight-distribution with parameters such as Gaussian distribution or exponential family instead of using MC estimator to approximate (1). When a posterior has multi-modality or skew, they can give inaccurate approximations. Second, in many cases, sampling-free

BNN uses a method other than variational inference to obtain a posterior because evidence lower bound (ELBO) is not amenable. These constraints make it difficult to apply sampling-free BNNs to deep NNs. Therefore, we mainly consider sampling-based BNN for comparison in this paper.

## 2.2 BAYESIAN NEURAL NETWORK GRADIENT ESTIMATION

It is standard to use a gradient of MC estimator to optimize the loss function of BNN, e.g. ELBO. Several direct and indirect methods are used to obtain the MC gradient estimation: reparametrization trick (Kingma & Welling, 2013; Blundell et al., 2015; Fortunato et al., 2017), score function (Kleijnen & Rubinstein, 1996; Williams, 1992; Glynn, 1990), dropout (Gal & Ghahramani, 2016; McClure & Kriegeskorte, 2016), batch normalization (Teye et al., 2018), expectation propagation (Hernández-Lobato & Adams, 2015), and smooth transformation for samples (Liu & Wang, 2016; Burda et al., 2015; Maddison et al., 2017; Naesseth et al., 2017; Le et al., 2017; van den Oord et al., 2017).

However, it is not appropriate to use these techniques to calculate a gradient of MC estimator $\mathcal{L} = \mathbb{E}_{p(\boldsymbol{x})}\big[f(\boldsymbol{x})\big]$ for arbitrary functions $f(\cdot)$ when a data stream is non-stationary and $p(\boldsymbol{x})$, the distribution of the data stream, is time-variant. For example, the reparameterization trick, one of the most widely used methods for MC gradient estimation, separates the probability $p(\boldsymbol{x})$ into a deterministic function $h(\cdot)$ and an invariant probability $p(\epsilon)$, i.e., $\boldsymbol{x} = h(\epsilon)$ and $\partial_{\boldsymbol{x}} p(\epsilon) = 0$. Then, the $\mathcal{L}$ is expectation over the invariant distribution, i.e., $\mathcal{L} = \mathbb{E}_{p(\epsilon)}\big[f(\boldsymbol{x})\big]$, and the approximated $\mathcal{L}$ becomes differentiable, i.e., $\partial_{\boldsymbol{x}} \mathcal{L} = \mathbb{E}_{p(\epsilon)}\big[\partial_{\boldsymbol{x}} f(\boldsymbol{x})\big]$. To use this technique, we need to adapt $h(\cdot)$ to the non-stationary data stream. If $h(\cdot)$ is modeled in an NN, we may need to continuously train this NN on the data stream; this leads to a decrease in computational performance. To alleviate this problem, we use histogram to estimate probability with nearest neighbor search, instead of using an NN-based generative model. In addition, the histogram is a linear sum of disjoint bins that is analytically useful while simplifying calculations.

## 3 ONLINE CODEVECTOR HISTOGRAM

Vector quantization was introduced in Gersho (1982); Gray (1984) in order to compress a probability distribution to a dozens of samples called codevectors. Kotani et al. (2002) showed that the histogram of codevectors—codevectors augmented with counts—can effectively represent the features of face image dataset. Xu et al. (2012); Frezza-Buet (2014); Ghesmoune et al. (2016) have proposed algorithms that changes codevectors depending on the data stream. We introduce *online codevector histogram* (OCH) to estimate the probability distribution of non-stationary data stream as well as dataset. OCH is a probabilistic algorithm that changes not only codevectors but their counts and calculates the difference in distribution as the data stream changes, at low computational cost. OCH can add a new codevector and delete old ones, while counting the matches of each codevector for the past data stream. OCH maps the input vector to the codevector using nearest neighbor search. It is a high-dimensional histogram where the Voronoi diagram is the boundary and a codevector represents the corresponding bin. DBNN uses OCH to approximate the probability distributions of input and output vector data streams.

Algorithm 1 shows how OCH operates in three steps. *(a)* Given a new input vector, OCH finds the nearest codevector and increases its count. Then, it divides the corresponding bin by inserting the input vector as a new codevector with probability proportional to the count. *(b)* OCH decrease all counts by the same rate to reduce the contribution of old data. *(c)* To keep the number of codevectors small, OCH deletes codevectors with probability in inverse proportion to its counts.

OCH takes three hyperparameters: $K$, $\lambda$, and $\phi$. $K$ is the number of codevectors kept in OCH on average. $\lambda$ determines how fast OCH depreciates old counts with the rate of $\gamma$. $\phi$ with counts of each bin regulates the probability of adding and deleting codevectors.

Nearest neighbor search is the most computationally intensive step in alg. 1. To lower the computational complexity, we use locality-sensitive hashing with stable distribution, $h = \lfloor (\boldsymbol{a} \cdot \boldsymbol{x}_\star + b)/w \rceil$ for a vector $\boldsymbol{a}$ and scalars $b$ and $w$. It requires up to $\log K$ hashes for precise search. The upper bound of the computational complexity is $\mathcal{O}(\dim(\boldsymbol{x}) \log K)$. This is faster than continuous training of NN-based generative models for the data stream changing over time.

---

**Algorithm 1:** Update OCH

---

**input** : input vector $\boldsymbol{x}_\star$, count of input vector $n_\star$, OCH $= \{(\boldsymbol{c}_i, n_i)\}$ where $\boldsymbol{c}_i$ is codevector and $n_i$ is its count, hyperparameters $K$, $\lambda$, and $\phi$

**output :** updated OCH

1   $\boldsymbol{c}_i \leftarrow$ Search the nearest codevector to $\boldsymbol{x}_\star$ in OCH

2   $n_i \leftarrow n_i + n_\star$ where $n_i$ is count of $\boldsymbol{c}_i$

3   $p \sim \text{Bernoulli}\big(\sigma(\pi_i - \bar{\pi} + \phi)\big)$ where $\sigma(\cdot)$ is sigmoid, $N = \sum_i n_i$, $\pi_i = n_i/N$, and $\bar{\pi} = 1/K$

4   **if** $p = 1$ **then**

5      OCH $\leftarrow$ OCH $\cup \{(\boldsymbol{x}_\star, (1 - \gamma) \cdot n_i)\}$ where $\gamma = e^{-\lambda/N}$

6      $n_i \leftarrow \gamma \cdot n_i$

7   **forall** $(\boldsymbol{c}_j, n_j) \in$ OCH **do**

8      $n_j \leftarrow \gamma \cdot n_j$

9   **forall** $(\boldsymbol{c}_k, n_k) \in$ OCH **do**

10      $q \sim \text{Bernoulli}\big(\sigma(\bar{\pi} - \pi_k + \phi) \cdot \bar{\pi}\big)$

11      **if** $q = 1$ **then**

12         OCH $\leftarrow$ OCH $\setminus \{(\boldsymbol{c}_k, n_k)\}$

---

Meanwhile, OCH approximates probability distribution $p(\boldsymbol{x})$ as:

$$p(\boldsymbol{x}) \simeq \sum_i \pi_i \mathcal{V}(\boldsymbol{x}|\boldsymbol{c}_i) \tag{4}$$

where $N = \sum_i n_i$, $\pi_i = n_i/N$, and $\mathcal{V}(\boldsymbol{x}|\boldsymbol{c})$ is the bin or *neighborhood* of $\boldsymbol{c}$ with $\int \mathcal{V}(\boldsymbol{x}|\boldsymbol{c})d\boldsymbol{x} = 1$. The right-hand-side of (4) is clearly differentiable with respect to $\pi_j$, while a set of random samples from $p(\boldsymbol{x})$ is not differentiable. Given a new input data, OCH changes only one count of the nearest codevector. Therefore, if $n_\star \ll N$, the difference of OCH for a new input vector is approximately:

$$\delta p(\boldsymbol{x}) \simeq \sum_i \delta \pi_i \mathcal{V}(\boldsymbol{x}|\boldsymbol{c}_i) \tag{5}$$

$$\simeq \alpha \mathcal{V}(\boldsymbol{x}|\boldsymbol{c}_\star) \tag{6}$$

where $\boldsymbol{c}_\star$ is the nearest codevector to the input vector, $\alpha = {\delta n}/{N}$, and $\delta n$ is the difference of the count of $\boldsymbol{c}_\star$. When OCH creates a new codevector, $\delta n$ is $n_\star$. Here, we used the fact that codevectors are invariant for input data, so $\mathcal{V}(\boldsymbol{x}|\boldsymbol{c}_i)$ is also invariant. This property is useful when the linear operator $g(\cdot)$ is applied to $p(\boldsymbol{x})$. By definition of linear operator, $g(p(\boldsymbol{x})) = \sum_i \pi_i g(\mathcal{V}(\boldsymbol{x}|\boldsymbol{c}_i))$, then its difference is $\delta g(p(\boldsymbol{x})) = \sum_i \delta \pi_i g(\mathcal{V}(\boldsymbol{x}|\boldsymbol{c}_i)) = \alpha g(\mathcal{V}(\boldsymbol{x}|\boldsymbol{c}_\star))$.

## 4   Differentiable Bayesian Neural Network Inference

When a data stream $\mathcal{S} = \{\dots, \boldsymbol{x}_\star\}$ is given, where each subsequent data sample changes continuously, the prediction of a NN for $\mathcal{S}$ also changes continuously, i.e., $p(\boldsymbol{y}) \to p(\boldsymbol{y}) + \delta p(\boldsymbol{y})$, because a NN (with continuous activation) is a homeomorphism. We show that $\delta p(\boldsymbol{y})$ is approximately proportional to the prediction of a NN augmented with OCH to the input and output for the recent data $\boldsymbol{x}_\star$. Figure 1 shows the structure of DBNN, where DNN is augmented with OCHs as distribution estimator units of input and output stream. DBNN calculates predictive uncertainty as well as the predictive result using the weighted ensemble of output codevectors.

### 4.1   Differentiation of Bayesian Neural Network Inference as Difference of Prediction

It is challenging to calculate the difference of (1) for a new data $\boldsymbol{x}_\star$ from $\mathcal{S}$. The differentiation of $p(\boldsymbol{y}|\boldsymbol{x}_\star, \boldsymbol{w})$ with respect to $\boldsymbol{x}_\star$ is analytically intractable, because the samples from the probability of the data stream $p(\boldsymbol{x}|\mathcal{S})$ are discrete, although $p(\boldsymbol{x}|\mathcal{S})$ varies continuously. In other words, $\partial_{\boldsymbol{x}_\star} p(\boldsymbol{y}|\boldsymbol{x}_\star, \boldsymbol{w}) = \int p(\boldsymbol{y}|\boldsymbol{x}, \boldsymbol{w}) \partial_{\boldsymbol{x}_\star} \delta(\boldsymbol{x} - \boldsymbol{x}_\star) d\boldsymbol{x}$ where $\delta(\cdot)$ is the delta function, and the delta function is not differentiable. To address this issue, DBNN smoothens the delta function of the data

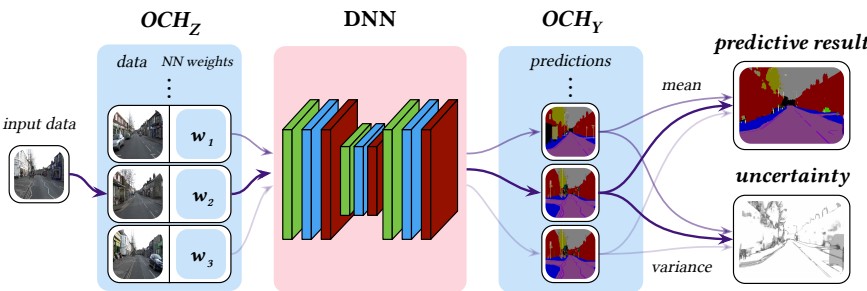

Figure 1: DBNN Inference. OCHs are added to the DNN to estimate the probabilities of input and output vector streams. Given a new data, $\text{OCH}_{\boldsymbol{Z}}$ adjusts the weight (arrows) of the nearest codevector of $\text{OCH}_{\boldsymbol{Z}}$ representing the input vector stream. DBNN adjusts weights of the codevectors of the $\text{OCH}_{\boldsymbol{Y}}$ representing the predictive distribution based on the results of the inner DNN. DBNN derives predictive result and uncertainty from the weighted ensemble of the codevectors of $\text{OCH}_{\boldsymbol{Y}}$.

sample $\delta(\boldsymbol{x} - \boldsymbol{x}_\star)$ to the probability of the data stream $p(\boldsymbol{x}|\mathcal{S})$ as follows:

$$p(\boldsymbol{y}|\mathcal{S}) = \int p(\boldsymbol{y}|\boldsymbol{x}, \boldsymbol{w})p(\boldsymbol{x}|\mathcal{S})p(\boldsymbol{w})d\boldsymbol{x}d\boldsymbol{w} \tag{7}$$

$$= \int p(\boldsymbol{y}|\boldsymbol{z})p(\boldsymbol{z}|\mathcal{S})d\boldsymbol{z} \tag{8}$$

where $\boldsymbol{z} = (\boldsymbol{x}, \boldsymbol{w})$ and $p(\boldsymbol{z}|\mathcal{S}) = p(\boldsymbol{x}|\mathcal{S})p(\boldsymbol{w})$. $p(\boldsymbol{x}|\mathcal{S})$ is separately learned with respect to the data stream $\mathcal{S}$ by another model and is called *data uncertainty*. $p(\boldsymbol{w})$ is the posterior distribution obtained from BNN training and is called *model uncertainty*. DBNN inference decouples data (stream) from the model. Instead, it joins the probability of data and the probability of the NN weights. When the most recent data sample is considered instead of the data stream, i.e., $p(\boldsymbol{x}|\mathcal{S}) = \delta(\boldsymbol{x} - \boldsymbol{x}_\star)$, DBNN inference is reduced to BNN inference as mentioned before.

To calculate (8), we use OCH and (4) to represent $p(\boldsymbol{z}|\mathcal{S})$, and call it $\text{OCH}_{\boldsymbol{Z}}$:

$$p(\boldsymbol{y}|\mathcal{S}) \simeq \sum_i \pi_i \int p(\boldsymbol{y}|\boldsymbol{z})\mathcal{V}(\boldsymbol{z}|\boldsymbol{c}_i)d\boldsymbol{z} \tag{9}$$

where $\pi_i$ is weight of codevector $\boldsymbol{c}_i$ and proportional to the count of the codevector. $\mathcal{V}(\boldsymbol{z}|\boldsymbol{c}_i)$ is neighborhood of $\boldsymbol{c}_i$. Then, $\delta p(\boldsymbol{y}|\mathcal{S})$ can be approximated by changing the weight of the codevector nearest to $\boldsymbol{x}_\star$ as:

$$\delta p(\boldsymbol{y}|\mathcal{S}) \simeq \sum_i \delta\pi_i \int p(\boldsymbol{y}|\boldsymbol{z})\mathcal{V}(\boldsymbol{z}|\boldsymbol{c}_i)d\boldsymbol{z} \tag{10}$$

$$\simeq \alpha \int p(\boldsymbol{y}|\boldsymbol{z})\mathcal{V}(\boldsymbol{z}|\boldsymbol{c}_\star)d\boldsymbol{z} \tag{11}$$

according to the (6). This means that $\int p(\boldsymbol{y}|\boldsymbol{z})\mathcal{V}(\boldsymbol{z}|\boldsymbol{c}_i)d\boldsymbol{z}$ is invariant with respect to the change of $\mathcal{S}$, i.e., $\delta\big[\int p(\boldsymbol{y}|\boldsymbol{z})\mathcal{V}(\boldsymbol{z}|\boldsymbol{c}_i)d\boldsymbol{z}\big] = 0$, since histogram only changes its counts, not the codevectors. We approximate $\mathcal{V}(\boldsymbol{z}|\boldsymbol{c}_\star)$ as delta-function distribution at $\boldsymbol{c}_\star$, i.e., $\mathcal{V}(\boldsymbol{z}|\boldsymbol{c}_\star) \simeq \delta(\boldsymbol{z} - \boldsymbol{c}_\star)$:

$$\delta p(\boldsymbol{y}|\mathcal{S}) \simeq \alpha p(\boldsymbol{y}|\boldsymbol{c}_\star) \tag{12}$$

It is safe to assume that $p(\boldsymbol{y}|\boldsymbol{c}_\star)$ is dominantly distributed near $\text{NN}(\boldsymbol{c}_\star)$, i.e., $p(\boldsymbol{y}|\boldsymbol{c}_\star) \simeq \mathcal{V}\big(\boldsymbol{y}|\text{NN}(\boldsymbol{c}_\star)\big)$, because the expected value of $\boldsymbol{y}$ should be equal to $\text{NN}(\boldsymbol{c}_\star)$:

$$\delta p(\boldsymbol{y}|\mathcal{S}) \simeq \alpha \mathcal{V}\big(\boldsymbol{y}|\text{NN}(\boldsymbol{c}_\star)\big) \tag{13}$$

where $\text{NN}(\boldsymbol{c}_\star)$ is prediction of DNN for $\boldsymbol{c}_\star$. This result is equal to the changes of OCH when we update OCH for data, as (6) expresses. In conclusion, the difference of the DBNN prediction contributes only to the neighborhood of the DNN prediction for the codevector in $\text{OCH}_{\boldsymbol{Z}}$ approximately. Thus, given the $\text{OCH}_{\boldsymbol{Y}}$ representing $p(\boldsymbol{y}|\mathcal{S})$, $p(\boldsymbol{y}|\mathcal{S}) + \delta p(\boldsymbol{y}|\mathcal{S})$ can be approximated by updating the $\text{OCH}_{\boldsymbol{Y}}$ for the prediction of DNN.

---

**Algorithm 2:** DBNN Inference

---

**input** : input data vector $\boldsymbol{x}_\star$, deterministic neural network NN$(\cdot)$, posterior distribution OCH$_{\boldsymbol{W}}$, distribution of input vector OCH$_{\boldsymbol{Z}}$, distribution of output vector OCH$_{\boldsymbol{Y}}$, cache table $T$ initialized to empty set

**output** : updated distribution of input vector OCH$_{\boldsymbol{Z}}$, updated distribution of output vector OCH$_{\boldsymbol{Y}}$

1  $\boldsymbol{w}_\star \sim \text{OCH}_{\boldsymbol{W}}$
2  $\boldsymbol{z}_\star \leftarrow (\boldsymbol{x}_\star, \boldsymbol{w}_\star)$
3  OCH$_{\boldsymbol{Z}} \leftarrow$ Update OCH$_{\boldsymbol{Z}}$ for $\boldsymbol{z}_\star$
4  **if** new codevector $\boldsymbol{c}_i$ exists in OCH$_{\boldsymbol{Z}}$ **then**
5  $\quad\mid\quad \boldsymbol{y}_i = \text{NN}(\boldsymbol{c}_i)$
6  $\quad\mid\quad T \leftarrow T \cup \{\boldsymbol{c}_i \mapsto \boldsymbol{y}_i\}$
7  $\boldsymbol{c}_\star \leftarrow$ Search the nearest neighbor codevector to $\boldsymbol{x}_\star$ in OCH$_{\boldsymbol{Z}}$
8  $\boldsymbol{y}_\star \leftarrow T(\boldsymbol{c}_\star)$
9  OCH$_{\boldsymbol{Y}} \leftarrow$ Update OCH$_{\boldsymbol{Y}}$ for $\boldsymbol{y}_\star$ with count $\alpha$

---

### 4.2 IMPLEMENTATION OF DIFFERENTIABLE BAYESIAN NEURAL NETWORK INFERENCE

DBNN is composed of three stages and Algorithm 2 describes the DBNN inference process in detail as follows: *(a)* First, to estimate the probability of the input data stream, the algorithm updates OCH$_{\boldsymbol{Z}}$ for $\boldsymbol{z}_\star = (\boldsymbol{x}_\star, \boldsymbol{w}_\star)$ where $\boldsymbol{x}_\star$ is an input data and $\boldsymbol{w}_\star$ is a random sample from given approximated posterior distribution OCH$_{\boldsymbol{Z}}$ which represents the given $p(\boldsymbol{w})$. OCH$_{\boldsymbol{W}}$ consists of weights sampled from trained $p(\boldsymbol{w})$ using MCMC. OCH$_{\boldsymbol{Z}}$ and OCH$_{\boldsymbol{Y}}$ are initialized to empty sets. *(b)* Second, if OCH$_{\boldsymbol{Z}}$ generated a new codevector in OCH$_{\boldsymbol{Z}}$, DBNN generates the prediction for the new codevector using DNN and keeps the prediction in a cache table. As a result, the cache table contains the results of DNN corresponding to all codevectors in OCH$_{\boldsymbol{Z}}$. *(c)* Third, DBNN finds the nearest neighbor codevector to $\boldsymbol{x}_\star$ in OCH$_{\boldsymbol{Z}}$, looks up the corresponding prediction in the cache table and update OCH$_{\boldsymbol{Y}}$ for the prediction for the codevector. In conclusion, OCH$_{\boldsymbol{Z}}$ approximating $p(\boldsymbol{z}|\mathcal{S})$ consists of recent data and weights from posterior. OCH$_{\boldsymbol{Y}}$ approximating $p(\boldsymbol{y}|\mathcal{S})$ consists of recent predictions.

DBNN calculates the difference of prediction for a new data sample from the previous predictions. In the process, it executes DNN once if necessary, which is computationally expensive, in contrast to BNNs' repetitive execution DNNs. Furthermore, DBNN sometimes does not execute DNN, but only updates OCH$_{\boldsymbol{Z}}$ and OCH$_{\boldsymbol{Y}}$ for the input and output vectors, by updating the counts and estimates prediction and uncertainty based on cached results. The dominant part of the computational cost of updating OCH$_{\boldsymbol{Z}}$ is the inner product of a given input vector, i.e., $\boldsymbol{a} \cdot \boldsymbol{z}$, for nearest neighbor search. Given $\boldsymbol{z} = (\boldsymbol{x}, \boldsymbol{w})$ and $\boldsymbol{a} = (\boldsymbol{a}_0, \boldsymbol{a}_1)$, $\boldsymbol{a} \cdot \boldsymbol{z} = \boldsymbol{a}_0 \cdot \boldsymbol{x} + \boldsymbol{a}_1 \cdot \boldsymbol{w}$ holds. The codevector samples $\boldsymbol{w}$ from OCH$_{\boldsymbol{W}}$ is fixed and finite, and DBNN caches all $\boldsymbol{a}_1 \cdot \boldsymbol{w}$. Therefore, the average computational cost on $\boldsymbol{a}_1 \cdot \boldsymbol{w}$ is minuscule, when updating OCH$_{\boldsymbol{Z}}$. In conclusion, the upper bound of the computational complexity of DBNN inference is $\mathcal{O}(\dim(\boldsymbol{x}) \log K_{\boldsymbol{Z}} + \text{NN}(\cdot) + \dim(\boldsymbol{y}) \log K_{\boldsymbol{Y}})$ where $\mathcal{O}(\text{NN}(\cdot))$ is computational complexity of NN and $K_{\boldsymbol{Z}}$ and $K_{\boldsymbol{Y}}$ are hyperparameters $K$ of OCH$_{\boldsymbol{Z}}$ and OCH$_{\boldsymbol{Y}}$, respectively. Modern deep NN performs vector operations dozens of times, if not hundreds, so the computation time of OCH is very small compared to DNN execution.

DBNN uses the flexible parametric probability estimator OCH to represent the distributions of input and output vector streams. Unlike the most BNNs that depend on a parameterized model described in (2), DBNN does not depend on the specific model. OCH represents both continuous and discrete vector spaces, and so does DBNN. If DNN and posterior are given, we can easily convert them to DBNN without significant modifications: just add OCH to the input and output of a DNN to estimate the probability of the input and output vector streams.

## 5 EXPERIMENTS

Although DBNN uses assumptions that seem reasonable, it is necessary to measure empirical performances to show that this assumption works for real-world problems. This section evaluates the performance of DBNN in three set of experiments. The first experiment visualizes the characteristics of DBNN by performing simple linear regression on synthetic data. The second experiment classifies various real-world multivariate datasets using shallow and narrow NNs. This experiment shows the

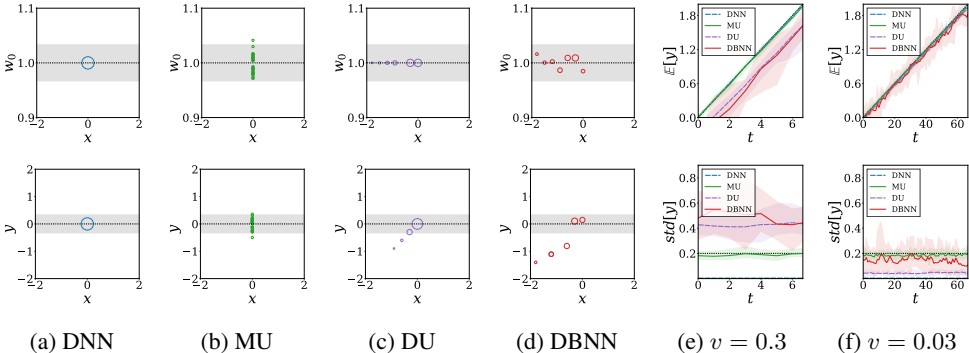

(a) DNN      (b) MU      (c) DU      (d) DBNN      (e) $v = 0.3$      (f) $v = 0.03$

Figure 2: Simple linear regression on synthetic dataset. In figs. 2a to 2d, the top is the approximated input probability $p(x, w_0)$ and the bottom is the approximated output probability with data $p(x, y)$ at $t = 0$ with $v = 0.3$. In these figures, fig. 2a and fig. 2b are samples from the distribution, and fig. 2c and fig. 2d are distributions approximated by $OCH_Z$ and $OCH_Y$. The sizes of the circles represents the importance weights of each codevector. Figures 2e to 2f are the expected values and standard deviations of $y$ depending on the timestamp for different $v$. The black dotted lines represent true values. The error is 90% confidence interval.

difference between computational and predictive performance of BNN and DBNN under various conditions when converting BNN to DBNN using small NN. The third experiment performs semantic segmentation on real-world video sequences. This experiment compares the performances of DBNN with other baselines when using a deep and wide modern NNs in practical situation.

We use the following four methods as baselines:

- **DNN.** Let $\text{Softmax}(\boldsymbol{y}')$ be probability of NN where $\boldsymbol{y}'$ is DNN logits. It is easy to implement, but it differs from the actual classification probability when the NN is deepened, broadened, and regularized well (Guo et al., 2017).

- **Model Uncertainty (MU).** BNN is referred to as MU in this section since it introduces model uncertainty to NN. MU have to calculate (3) to predict the result. It is difficult to analytically determine the number of samples that the prediction converges. Instead, in section 5.1 and section 5.2, we experimentally set the number of samples from posterior to 30 so that the accuracy and the negative log-likelihood (NLL) converge. In section 5.3, MU containing MC dropout (Gal & Ghahramani, 2016) layers predicts results with 30 forward passes. Appendix E shows the performance of the MU for different numbers of forward passes.

- **Data Uncertainty (DU).** DU is a method that assigns only data uncertainty to DNN by adding OCHs to input and output of trained DNN. In other words, it uses one weight instead of weight distribution in the DBNN, i.e., (8) with $p(\boldsymbol{w}) = \delta(\boldsymbol{w} - \boldsymbol{w}_\star)$. This method shows the effect of OCH on prediction when it is applied to the NN. Also, it shows that DNNs modified using OCH can achieve higher uncertainty than vanilla DNNs in semantic segmentation experiment.

- **DBNN.** DBNN takes both data uncertainty and model uncertainty. DBNN adds OCH to the input and output of trained BNN. In section 5.1 and section 5.2, the approximated posterior distribution of this method $OCH_W$ consists of 30 samples from the BNN's posterior.

## 5.1 SIMPLE LINEAR REGRESSION

A neural network with a linear activation function is a linear regression model $y = w_0 x + w_1$. The posterior is given by $p(w_0) = \mathcal{N}(1.0, 0.02^2)$ and $p(w_1) = \mathcal{N}(0.0, 0.2^2)$. The distribution of time-varying input data streams is given by $p(x|t) = \delta(x - vt)$ where $t$ is integer timestamp from $t = -10$. $v$ is 0.3 and 0.03. The top of figs. 2a to 2d are approximated distributions of input and weight samples $p(x, w_0)$ and bottom are approximated distributions of output samples with data $p(x, y)$ at $t = 0$. $w_1$ is omitted from $\boldsymbol{w}$ in these figures, but it behaves like $w_0$.

Table 1: Predictive performance of MU and DBNN for classification on multivariate datasets.

| Dataset | RMSE | | RMSE-90 | | NLL | | Cov-90 | |
|---------|------|------|---------|------|------|------|--------|------|
| | MU | DBNN | MU | DBNN | MU | DBNN | MU | DBNN |
| Localization | 0.228 | 0.212 | 0.118 | 0.085 | 1.252 | 1.084 | 13.1 | 3.11 |
| EMG | 0.260 | 0.249 | 0.204 | 0.203 | 0.955 | 0.935 | 10.8 | 9.91 |
| Occupancy | 0.134 | 0.152 | 0.071 | 0.076 | 0.071 | 0.081 | 90.1 | 90.1 |

To predict result, DNN uses point-estimated weight and data $x = 0$, which is the data at $t = 0$. MU uses weight distribution, instead of point-estimated weight, and data $x = 0$. DU uses point-estimated weight, which is the same as DNN. However, since DU contains OCHs, it uses not only the most recent data ($x = 0$ at $t = 0$) but also past data ($x < 0$ when $t < 0$). As a result, DU predicts the distribution of $y$. DBNN predicts the distribution of $y$ using both weight distribution and data from the past to now. DBNN without distribution of $\boldsymbol{x}$ is equivalent to MU, and DBNN without distribution of $\boldsymbol{w}$ is equivalent to DU.

The figs. 2e to 2f show expected value and standard deviation of DBNN prediction on time. First, as shown in top of fig. 2e, the result of DBNN lags behind since DBNN smoothens its prediction with respect to time.[1] Second, unlike MU uses dozens of samples, DBNN uses a small number of samples, so the error is relatively large. Third, as shown in bottom of fig. 2e, the DBNN prediction is under-confident because DBNN considers not only MU but also DU. Fourth, as shown in the comparison of fig. 2e and fig. 2f, DBNN converges to BNN and DU converges to DNN as the input data stream changes more slowly.

## 5.2 CLASSIFICATION ON MULTIVARIATE DATASETS

A classification experiment is designed to compare computational and predictive performance changes when converting BNNs to DBNNs using shallow and narrow NNs in various situations. In this experiment, we use three time-series real-world datasets with input dimensions between 4 and 8. Localization and EMG dataset have relatively large number of classes, eleven and eight respectively while Occupancy dataset has only two classes. The NNs consist of 2 fully-connected hidden layers with 50 units. The distributions of weights are optimized using variational inference. See appendix A for more information about experimental settings and datasets.

**Computational Performance.** Execution times for one batch of MU and DBNN are 26.7±10.6ms and 19.5±10.0ms, respectively. MU parallelizes execution by using a batch size of 30. This result shows that DBNN is 37% faster than MU even though MU is parallelized and DBNN uses additional OCHs. DBNN improves the computational performance of DBNN by sampling the weights from the posterior before testing. As NN gets deeper and larger, the execution time of the DBNN will be reduced compared to the MU, because the larger the NN, the greater the burden of sampling weights.[2]

**Predictive Performance.** As shown in Table 1, we compared MU and DBNN in terms of the root-mean-square error (RMSE), RMSE for predictive results with confidence above 90% (RMSE-90), negative log-likelihood (NLL), and the percentage of predictive results with confidence above 90% (Cov-90) of the MU and DBNN for three datasets. DBNN is more accurate than MU and predicts uncertainty better than MU on Localization and EMG datasets. DBNN predicts less accurately than MU on Occupancy dataset with comparable uncertainties. The confidence of DBNN is always less than or equal to that of MU. Overall, DBNN provides predictive performance comparable to BNN on various datasets.

---

[1] The delay of the expected value of $\boldsymbol{y}$ over time does not mean that the regression result is biased. DBNN estimates the correct value on average since $\boldsymbol{z}$ is also delayed just as $\boldsymbol{y}$ is delayed.

[2] Execution times of MU and DBNN with 10 fully-connected hidden layer NN are 98.9±20.1ms and 49.0±23.6ms, respectively. In this case, DBNN is 2.02× faster than MU. See appendix B for more information about the relationship between NN depth and execution time.

Table 2: Computational and predictive performance with semantic segmentation for each method.

| Method | Thr (fps) | Acc | Acc-90 | Unc-90 | IoU | IoU-90 | NLL | Cov-90 |
|--------|-----------|------|--------|--------|------|--------|-------|--------|
| DNN | 6.14 | 85.8 | 89.1 | 30.4 | 58.5 | 62.5 | 1.22 | 93.1 |
| MU | 0.189 | 86.4 | 93.0 | 60.1 | 61.0 | 69.9 | 0.728 | 84.2 |
| DU | 5.33 | 85.4 | 91.5 | 51.3 | 57.3 | 63.3 | 0.980 | 86.0 |
| DBNN | 5.22 | 85.8 | 92.3 | 63.0 | 58.9 | 68.6 | 0.826 | 80.4 |

## 5.3 SEMANTIC SEGMENTATION

Semantic segmentation experiment shows the computational and predictive performance of DBNN with a modern deep NN in practical situation. We use CamVid dataset (Brostow et al., 2009) consisting of 30 frame-per-second (fps) video sequences of real-world day and dusk road scenes. We use the U-Net (Ronneberger et al., 2015) as the backbone architecture. Bayesian U-Net, similar to Kendall et al. (2015), contains MC dropout (Gal & Ghahramani, 2016) layers. For more information about experimental settings, see appendix A.

**Computational Performance.** The throughput (Thr) column of table 2 shows the number of video frames processed by each model per second. This table shows that DNN takes $162\pm12$ms on average to process one frame. In comparison, DBNN takes $191\pm70$ms on average to process one frame, which is only 17% higher than DNN. According to the difference between the execution time of DNN and DU, the average execution time of one OCH is 14ms, which is only 7% of the total. Besides, due to GPU memory limitations, the batch size of MU is limited to 1. Thus, the MU predicts results for 30 batches of size 1. In conclusion, the execution time of MU is $5291\pm69$ms, which is $32\times$ higher than that of DNN, and $27\times$ higher than that of DBNN. Moreover, DBNN can increase throughput at the expense of accuracy by adjusting a hyperparameter. See appendix C for more information.

**Predictive Performance.** The Acc to Cov-90 columns of the table 2 show the quantitative comparison of the predictive performance for each method. We measure global pixel accuracy (Acc), mean intersection over unit (IoU), and negative log-likelihood (NLL). At the same time, we selected only those pixels with confidence greater than 90% and measure the accuracy (Acc-90), i.e., $p(\textbf{accurate}|\textbf{confident})$, and IoU (IoU-90). We also measure the probability that the pixel is not 90% confident pixel when prediction is incorrect (Unc-90), i.e., $p(\textbf{unconfident}|\textbf{inaccurate})$, and the percentage of pixels with 90% confidence (Cov-90).

According to this table, the accuracies of DU and DBNN are 0.46% and 0.69% lower than that of DNN and MU, respectively. For pixels with ninety-percent or higher confidence, DNN, MU, DU, and DBNN predicts with 3.8%, 7.6%, 7.1%, and 7.5% higher accuracy, respectively, compared to the accuracy for all pixels. MU, DU and DBNN improves the accuracy more than DNN for confident pixels, which means that MU and DBNN estimates high uncertainty for misclassified region. IoUs for certain pixels increase by 6.8%, 14%, 10%, 16%, respectively, compared to the IoUs for all pixels, which shows the same trend as in accuracy.

As shown in this results, DNN is the most improper way to distinguish uncertain pixels because it has the least performance improvement compared to other methods for pixels with high confidence. On the other hand, MU has the highest performance improvement compared to DNN when considering all pixels, and it also has the highest performance for certain pixels. DBNN has improved performance for certain pixels, similar to MU. The predictive performance of DU for all pixels is similar to the predictive performance of DNN, but for certain pixels, the performance is much better than that of DNN. See appendix D for more information on semantic segmentation experiment. With a small number of forward passes, BNN can reduce computational overhead at the expense of predictive performance. To achieve equivalent performance to DBNN in uncertainty measures, MU requires 10 forward passes. See appendix E for more details.

## 6 CONCLUSION

We present a differentiable BNN (DBNN) inference with respect to input data, which is a novel approximation of BNN inference, to improve the computational performance of BNN inference for data streams. The derivative of DBNN predictive inference with respect to input data derives the increment of prediction when one data is newly given from the data stream. However, the inference of vanilla BNN cannot be differentiated with respect to data. To address this issue, DBNN introduce a new term that is the probability of data streams in the BNN inference. Then, it approximate its prediction with a histogram for high-dimensional vector streams. Consequently, the DBNN inference executes DNN only once to calculate the prediction changed by a new data from data streams. This results in an order of magnitude times improvement in computational performance compared to deep BNN. Experiments with semantic segmentation using real-world datasets show that the computational performance of DBNN is almost the same as that of DNN, and uncertainty is comparable to that of BNN.

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

Table 3: Input dimensionalities ($\dim(\boldsymbol{x})$), output dimensionalities ($\dim(\boldsymbol{y})$), and number of training sets ($N$) of dataset used in the experiments.

| Dataset | $\dim(\boldsymbol{x})$ | $\dim(\boldsymbol{y})$ | $N$ |
|---|---|---|---|
| Localization | 4 | 11 | 148373 |
| Occupancy | 5 | 2 | 8143 |
| EMG | 8 | 8 | 3793345 |
| CamVid | $720{\times}960{\times}3$ | $720{\times}960{\times}32$ | 421 |
| CityScape | $512{\times}1024{\times}3$ | $512{\times}1024{\times}35$ | 2975 |

## A  EXPERIMENTAL SETUP AND DATASETS

We conduct all the experiments with the Intel Xeon W-2123 Processor, 32GB memory, and a single GeForce RTX 2080 Ti. NN models are implemented in TensorFlow (Abadi et al., 2016).[3]

**Classification on Multivariate Datasets.**  In section 5.2, we train BNNs with Adam with a constant learning rate of 0.001 using ELBO loss. We set the batch size to 30. The BNN consist of 2 fully-connected layers with 50 units using Flipout estimator (Wen et al., 2018). We set hyperparameters $K$, $\lambda$, and $\sigma(\phi)$ to 10, 1.3, and 1.0, respectively, in all OCHs of DBNN; this is the same configuration in the semantic segmentation experiment.

We use the following time-sequential real-world dataset for classification in section 5.2: Localization Data for Person Activity Dataset (Kaluža et al., 2010), Occupancy Detection Dataset (Candanedo & Feldheim, 2016), and EMG Data for Gestures Dataset (Lobov et al., 2018). If there are no distinction between the test set and the training set, NNs use 90% of the dataset for training and the rest for testing. NNs test the datasets in time-sequential order. See table 3 for more information about datasets.

**Semantic Segmentation.**  In section 5.3 and appendix D, NNs are trained using categorical cross-entropy loss, with RMSprop with a constant learning rate of 0.0001 and discounting factor of 0.995. Batch size is limited to 1 because of memory limitations. In these experiments, we use U-Net (Ronneberger et al., 2015) based on VGG-16 (Simonyan & Zisserman, 2014). Bayesian U-Net, similar to Kendall et al. (2015), contains six MC dropout (Gal & Ghahramani, 2016) layers behind the layers that receives the smallest input vector sizes. The overhead of dropout layers are negligible and DBNN always uses a new dropout every time it predicts a result. To optimize the predictive performances, we set hyperparameters $K$, $\lambda$, and $\sigma(\phi)$ to 10, 1.3, and 1.0, respectively, in all OCHs of DU and DBNN. See appendix C for the change in NLL with hyperparameters. To improve computational performance, $\text{OCH}_{Y}$ executes nearest neighbor search for $\arg\max$ of $\boldsymbol{y}$ with respect to the classes.

We use CamVid dataset (Brostow et al., 2009) and CityScape dataset (Cordts et al., 2016) in the semantic segmentation experiments. We feed sequential images to DBNN and DU before evaluation since DU and DBNN use multiple previous images. For the CamVid dataset, we resized images to $720{\times}960$ pixel bilinearly from $360{\times}480$ pixels. Since some test sets do not have previous sequences, we use only 70 of 168 test sets in the CamVid dataset. For CityScape dataset, we use validation set, which was not used during training, as test set because only the video sequence corresponding to the validation set has been disclosed. See table 3 for more information about datasets.

## B  COMPUTATIONAL PERFORMANCE FOR NEURAL NETWORK DEPTH

Figure 3 shows a trend that the execution time of MU and DBNN increases as hidden layers get deeper without changing the input and output dimensions. In this case, as in section 5.2, NN consists of 50 units of fully-connected hidden layers, with input and output dimensions of 4 and 11, respectively. MU and DBNN use batches of size 30 and 1, respectively, but execution time is the same even if MU uses batch size of 1. According to this figure, when the number of hidden layers is 0, the execution time of DBNN is 15.1±6.7ms, which is 4.6ms slower than MU. This is because DBNN uses two additional OCHs compared to MU. However, unlike MU, DBNN does not sample the weight in the

---

[3]Code available at https://anonymous.4open.science/r/dbnn/

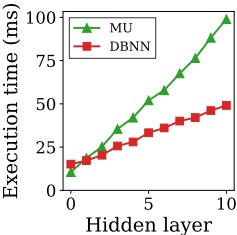

Figure 3: Prediction execution time of MU and DBNN for the number of hidden layers. The execution times increase linearly as the layer increases. If the number of hidden layers is 0, DBNN is slower than MU, but as the number of hidden layers increases, DBNN is faster than MU. The increase rates of execution times of MU and DBNN are 8.6ms/layer and 3.5ms/layer, respectively.

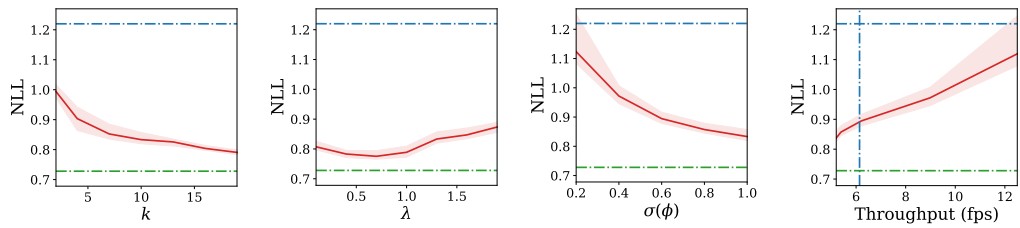

Figure 4: NLLs of DBNN for hyperparameters $K$, $\lambda$, and $\sigma(\phi)$, and the NLL for throughput (from left). In all figures, the NLLs of DNN and BNN are represented as blue and green dashdot horizontal lines, respectively. In the fourth figure, the throughput of DNN is represented as blue vertical dashdot line. The error is 80% confidence interval.

posterior when it predicts the result. Therefore, the increase of execution time of DBNN per layer is lower than that of MU. As a result, the execution time of the MU increases by 8.6ms while that of DBNN increases by 3.5ms when one layer is added. When the number of hidden layers is 10, the execution time of MU and DBNN is 98.9±20.1ms and 49.0±23.6ms, respectively—DBNN is 2.02× faster than MU.

## C  PREDICTIVE PERFORMANCE FOR HYPERPARAMETERS

DBNN has three hyperparameters; $K$, $\lambda$, and $\sigma(\phi)$, in the input and output OCHs. The first to third figures in fig. 4 shows how NLL changes over the variuos hyperparameters of the output OCH in semantic segmentation. As shown in this figure, as $K$ increases, NLL decreases, and the NLL stagnates when $K$ goes over 15. The lower the $\lambda$, the lower the NLL, until 0.7. The higher the $\sigma(\phi)$, the lower the NLL. The effect of hyperparameters of the input OCH is the similar trend as that of the output OCH.

With high $K$ and and low $\lambda$, DBNN maintains more codevectors—recent frames in video sequence and obtains more accurate model uncertainty using more codevectors. As a result, at high $K$ and low $\lambda$, DBNN exhibits lower NLL. When $K$ and $\lambda$ get lower than a certain point ($\lambda = 0.7$ in this example), however, the data uncertainty is too high to estimate results. NLL is minimized when model uncertainty and data uncertainty are balanced. On the other hand, if $\sigma(\phi)$ is low, DBNN mostly adjusts the weights of codevectors and occasionally adds new codevectors. This results in the DBNN becoming inaccurate, but the calculation is faster.

**Accuracy Throughput Trade-off.**  If $\sigma(\phi)$ is less than 1.0, DBNN occasionally adds a new codevector to the input OCH and executes NN, at the rate of $\sigma(\phi)$ on average. Since NN occupies the most of the execution time of DBNN, throughput decreases when $\sigma(\phi)$ decreases. The fourth figure in fig. 4 shows the changes in throughput and NLL as $\sigma(\phi)$ changes. According to this figure, DBNN can increase throughput up to 12fps when it achieves same NLL as DNN. In this case, DBNN is 2×

Table 4: Computational and predictive performance with semantic segmentation on the CityScape dataset.

| Method | Thr (fps) | Acc | Acc-90 | Unc-90 | IoU | IoU-90 | NLL | Cov-90 |
|--------|-----------|------|--------|--------|------|--------|-------|--------|
| DNN    | 7.09      | 89.3 | 93.8   | 55.3   | 61.3 | 72.6   | 0.569 | 89.7   |
| MU     | 0.236     | 89.2 | 95.9   | 74.8   | 61.5 | 79.3   | 0.382 | 84.2   |
| DU     | 5.88      | 87.5 | 94.3   | 67.5   | 56.9 | 69.8   | 0.567 | 82.8   |
| DBNN   | 5.89      | 87.2 | 95.4   | 76.8   | 56.4 | 72.0   | 0.452 | 78.2   |

faster than DNN. Conversely, DBNN can update a batch of two or more sizes instead of one. In this case, throughput of DBNN decreases but NLL increases (not shown in this figure).

# D    EXTENDED INFORMATIONS OF SEMANTIC SEGMENTATION EXPERIMENT

Section 5.3 shows the performance with semantic segmentation on the CamVid dataset. This section shows the performance on the CityScape dataset, another real-world road scene video sequence. Furthermore, we analyze the predictive performances of DNN, MU, DU, and DBNN in more detail.

**Predictive Performance of Models for Datasets.**    Table 4 shows the computational and predictive performance of various methods on the CityScape dataset. We measure the followings as in section 5.3: throughput (Thr), global pixel accuracy (Acc), accuracy for 90% certain pixels (Acc-90), the probability that the pixel is not 90% certain pixel when prediction is incorrect (Unc-90), mean intersection over unit (IoU), IoU for 90% certain pixels (IoU-90), negative log-likelihood (NLL), and the ratio of 90% certain pixels (Cov-90).

Table 4 implies the followings: First, The computational performance of DNN and DBNN are relatively comparable, and that of DBNN is significantly higher than that of MU. Second, the predictive performances of DU and DBNN for all pixels is lower than that of DNN and BNN, respectively. Third, predictive performances for certain pixels and other measures for uncertainties of MU and DBNN are comparable. These results are consistent with the result of section 5.3.

**Extended Analysis of Confidence and Accuracy.**    Figure 5 shows confidence histograms (Guo et al., 2017) and reliability diagrams (DeGroot & Fienberg, 1983; Niculescu-Mizil & Caruana, 2005; Naeini et al., 2015; Guo et al., 2017) of U-Net on the CamVid dataset. A confidence histogram shows proportion of samples for confidence as a histogram. The reliability diagram is a histogram of accuracy for confidence. We also measure the Expected Calibration Error (ECE) (Naeini et al., 2015; Guo et al., 2017), which is defined as follows for the reliability diagram:

$$\text{ECE} = \sum_i \frac{|B_i|}{M} \left| \text{acc}(B_i) - \text{conf}(B_i) \right| \tag{14}$$

where $|B_i|$ is the number of samples in bin $B_i$, $M = \sum_i |B_i|$, $\text{acc}(B_i)$ is mean accuracy of samples in $B_i$, and $\text{conf}(B_i)$ is mean confidence of samples in $B_i$. ECE stands for the difference between confidence and accuracy. ECEs of DNN, MU, DU, and DBNN are 7.43%, 4.27%, 5.86%, and 5.13%, respectively.

As shown in fig. 5a, DNN is overconfident and miscalibrated; its confidence is generally high and there is significant discrepancy between confidence and accuracy. On the other hand, MU is under-confident and well-calibrated. DBNN is more under-confident than MU, and the confidence of DBNN is calibrated relatively comparable to that of MU. Also, DU calibrates the confidence of DNN. In conclusion, the confidence of DBNN is well-calibrated, whild the confidence of the DNN is relatively unreliabl. These are consistent with the results of the predictive performance.

**Qualitative Results.**    Figure 6 shows the qualitative comparison of the predictions for each method. According to this figure, DNN is overconfident, i.e., confidence is generally high, and uncertainty is mostly distributed at the boundaries of the classified chunks. Even when DNN generates wrong prediction, the confidence level is very high. The uncertainty of MU is distributed on the boundaries

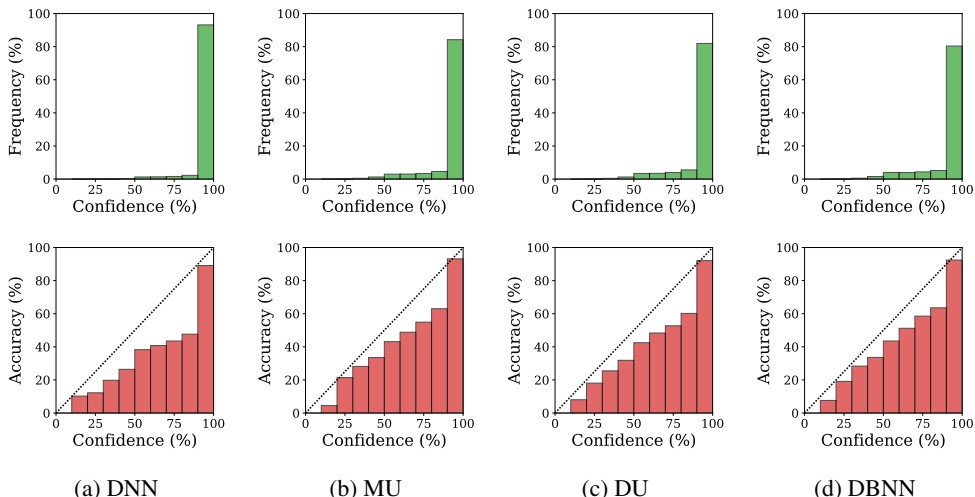

Figure 5: Confidence histograms and reliability diagrams of U-Net on the CamVid dataset. The top is the confidence histogram and the bottom is the the reliability histogram. The black dotted lines in the reliability diagram show the accuracy we expect for each confidence. There is a large gap between confidence and accuracy in the reliability diagram of the DNN, which means that it is poorly calibrated. On the other hand, MU is relatively well-calibrated.

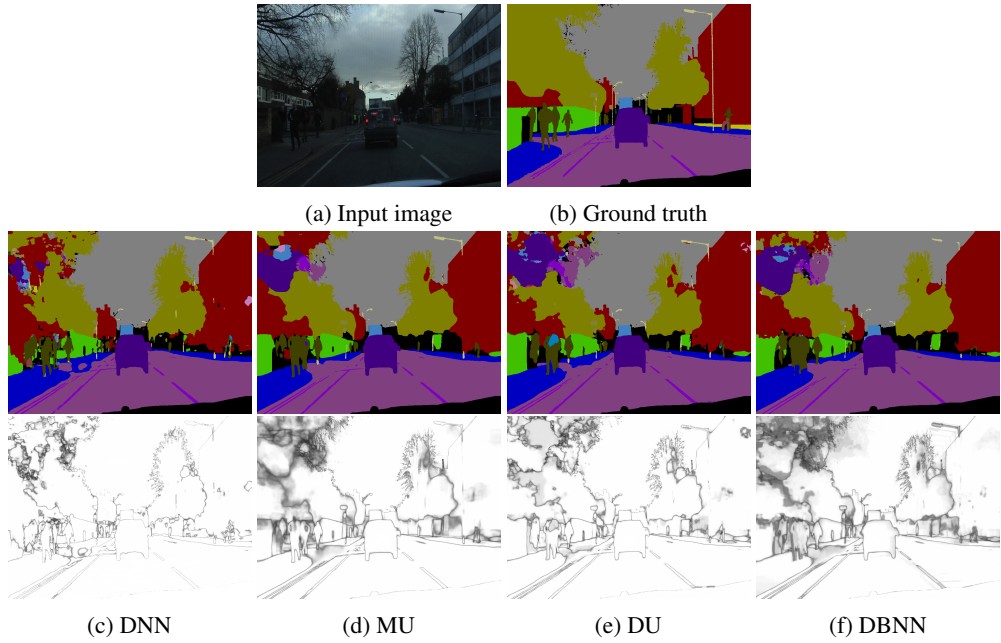

Figure 6: Qualitative results on the CamVid dataset for each method. In figs. 6c to 6f, the first row is the predictive result and the second row is the predictive confidence. A whiter background corresponds to higher confidence.

Table 5: Computational and predictive performance of U-Net with semantic segmentation for each method and number of forward passes ($N_y$) on CamVid dataset.

| Method | $N_y$ | Thr (fps) | Acc | Acc-90 | Unc-90 | IoU | IoU-90 | NLL | Cov-90 |
|--------|-------|-----------|------|--------|--------|------|--------|-------|--------|
| MU | 1 | 6.06 | 85.8 | 89.9 | 40.1 | 59.8 | 65.4 | 1.00 | 90.0 |
| | 2 | 2.97 | 86.1 | 91.3 | 50.7 | 60.3 | 67.6 | 0.892 | 87.0 |
| | 5 | 1.16 | 86.3 | 92.0 | 56.6 | 60.7 | 68.9 | 0.827 | 84.9 |
| | 10 | 0.580 | 86.4 | 92.4 | 59.5 | 60.9 | 69.6 | 0.768 | 84.3 |
| | 30 | 0.189 | 86.4 | 93.0 | 60.1 | 61.0 | 69.9 | 0.728 | 84.2 |
| | 50 | 0.115 | 86.4 | 93.0 | 60.3 | 61.0 | 70.1 | 0.721 | 84.2 |
| DBNN | 1.0 | 5.22 | 85.8 | 92.3 | 63.0 | 58.9 | 68.6 | 0.826 | 80.4 |

as in the case of the DNN, but is also distributed in the misclassified areas. The uncertainty of DU differs from that of DNN. First, DU is under-confident compared to DNN. Second, although DU does not identify all of the misclassifications compared to MU, it sometimes estimates high uncertainty in the misclassified areas. The uncertainty of DBNN is similar to the MU as we expected.

## E    PERFORMANCE OF BNN FOR NUMBER OF FORWARD PASSES

BNNs, i.e., MUs, in section 5.3 and appendix D contain MC dropout (Gal & Ghahramani, 2016) layers and require multiple forward passes to predict result. Table 5 shows the computational and predictive performance of the Bayesian U-Net for the number of forward passes on the CamVid dataset.

According to table 5, the predictive performance of MU improves up to 30 forward passes. However, there is little difference between the predictive performance for 30 forward passes and for 50 forward passes. MU requires at least 30 forward passes to get the best predictive performance, so we execute 30 predictive inference in section 5.3 and appendix D.

Meanwhile, to achieve the same uncertainty measures as DBNN, MU requires 10 forward passes. In particular, Unc-90 of DBNN has always outperformed that of MU. For the improvement rate from Acc and IoU to Acc-90 and IoU-90, the uncertainty of DBNN is the same as that of MU with 30 forward passes.

