# OpenReview forum: "Differentiable Bayesian Neural Network Inference for Data Streams"
_ICLR.cc/2020/Conference — Reject_

### Official Review · AnonReviewer1 · 2019-10-21
**Official Blind Review #1**

**Rating:** 3

**Review:**

Summary of the Paper:

This paper describes a method for training Bayesian neural networks in the context of stream data. The method proposed is based on using a quantization approach with some techniques to estimate the change in probability distributions. The proposed approach is compared on some tasks, including real and synthetic datasets.

Detailed comments:

The paper needs to improve the writing. For example, the sentence "it has been unable to estimate the uncertainty of the predictions until recently" sounds awkward.

The authors have to better explain the features and challenges of stream data.

The paper is unclear. There are several steps of the proposed method that are not well described.

How is the posterior distribution of the weights computed?

The notation x_0 and x_1 for the test point and the NN weights is confusing.

Figure 2 is not clear.

The description of the baselines the authors compare with is not clear.

What do you mean by degenerated in section 5.1.?

The paper is missing a related work section describing state of the art methods to address stream data.

It seems the real experiments of section 5.1. only consider one baseline MU. I believe this is insufficient.

In table 2 the benefits of the proposed approach are not very significant.

The experiments are missing error bars. It is not possible to extract conclusions of significance without them.

My overall impression is that the paper needs to better explain the approach followed and improve the notation to facilitate the reading. I believe that this paper needs for work and is not yet suitable for acceptance.


**Experience Assessment:**

I have read many papers in this area.

**Review Assessment: Checking Correctness Of Derivations And Theory:**

I did not assess the derivations or theory.

**Review Assessment: Checking Correctness Of Experiments:**

I did not assess the experiments.

**Review Assessment: Thoroughness In Paper Reading:**

I made a quick assessment of this paper.

---

> ### Author Response · Authors · 2019-11-11
> **Thank you for detailed and constructive comments.**
>
> We really appreciate your comments. We have revised the paper according to the suggestions and would like to clarify several things:
>
> 1. How is the posterior distribution of the weights computed?
>
> In this paper, as mentioned in the introduction and Section 4.2, we only consider predictive inference. We assume that we have access to a posterior. In the experiments, the posterior is given in Section 5.1. Section 5.2 trained the fully-bayesian NN using variational inference and Section 5.3 used BNN with MC-dropout.
>
>
> 2. Figure 2 is not clear. What do you mean by degenerated in Section 5.1.?
>
> "Degenerated" means the random variable has only one value, i.e., the distribution is a Dirac delta function. We removed the term "degenerated" from this revision to avoid confusion. Instead, we added detailed descriptions of Figure 2. Could you refer to reply to revewer 2's comment?
>
>
> 3. The description of the baselines the authors compare with is not clear.
>
> We compare four methods in the experiments. The first two methods are deterministic NN (DNN) and Bayesian NN (shown as MU). DU and DBNN are the methods that OCH is added to input and output of DNN and BNN, respectively. In short, DU = OCH + DNN + OCH and DBNN = OCH + BNN + OCH = posterior + OCH + DNN + OCH. We also added a description to this revision to clarify the details of the baselines.
>
>
> 4. The paper is missing a related work section describing state of the art methods to address stream data.
>
> We agree with you. In this revision, we cited some related work on vector quantization and data streams in Section 3.
>
>
> 5. In table 2 the benefits of the proposed approach are not very significant.
>
> Table 2 shows the computational and predictive performance of DBNN using modern NNs in semantic segmentation. In Table 2, we argue that DBNN is significantly faster than MU, and uncertainty is comparable in practical problem on real-world dataset. According to this table, the throughput of DBNN is significantly higher than that of MU. The uncertainty (measured by Acc-90, Unc-90, IoU-90, and NLL) of DBNN is comparable with that of MU, and is better estimated than that of DNN.
>
>
> 6. The experiments are missing error bars.
>
> Figure 2 has error bars.  As you pointed out, table 1 and table 2 have no error. Only the error of throughput is written in the Computational Performance paragraph. We didn't include the error numbers in table 1 and table 2, because we thought the table with errors is too wide and unreadable. For example, table 2 with errors is as follows:
>
>   Method | Thr (fps) | Acc | Acc-90 | Unc-90 | IoU | IoU-90 | NLL | Cov-90
>   —
>   DNN    | 6.14±0.43   | 85.8±0.0 | 89.1±0.0 | 30.4±0.0 | 58.5±0.0 | 62.5±0.0 | 1.22±0.00  | 93.1±0.0
>   MU       | 0.189±0.002 | 86.4±0.0 | 93.0±0.0 | 60.1±0.1 | 61.0±0.1 | 69.9±0.0 | 0.728±0.001 | 84.2±0.0
>   DU       | 5.33±1.36   | 85.4±0.1 | 91.5±0.2 | 51.3±0.7 | 57.3±0.2 | 63.3±0.3 | 0.980±0.010 | 81.9±0.4
>   DBNN  | 5.22±1.40   | 85.8±0.2 | 92.3±0.4 | 63.0±2.7 | 58.9±0.6 | 68.6±0.9 | 0.826±0.016 | 80.4±0.9
>
> We may add this expanded table to Appendix if it would be more informative.
>
>
> 7. It seems the real experiments of Section 5.1. only consider one baseline MU.
>
> Section 5.1 contains the results of DNN, MU, DU, and DBNN. If you are referring to table 1 in Section 5.2,  I would like to answer on the premise. A classification experiment in Section 5.2 is designed to compare performance changes when converting BNNs to DBNNs using shallow and narrow NNs in various situations. However, according to [Guo, 2017], shallow DNN is well-calibrated and predicts uncertainty well, and predictive results of DNN and BNN (not DBNN) are more dependent on the characteristics of data, not on the type of NN. Therefore, we excluded the results of DNN because they do not clearly show the purpose of this experiment.
>
> Reference:
> - Guo, Chuan, et al. "On calibration of modern neural networks." Proceedings of the 34th International Conference on Machine Learning-Volume 70. JMLR. org, 2017.
>
>
> 8. The paper needs to improve the writing. For example, the sentence "it has been unable to estimate the uncertainty of the predictions until recently" sounds awkward. The authors have to better explain the features and challenges of stream data. There are several steps of the proposed method that are not well described. The notation x_0 and x_1 for the test point and the NN weights is confusing.
>
> Thank you for your detailed comments. We added a more detailed explanation of the proposed method in this revision. In addition, we released the implementation of DBNN and experiments as open source. Readers will find more details about DBNN in the code. According to your suggestion, we changed the notation as follows to avoid confusion: data $\textbf{x}_0$ → $\textbf{x}$, weight $\textbf{x}_1$ → $\textbf{w}$, and $\textbf{x} = (\textbf{x}_0, \textbf{x}_1)$ → $\textbf{z} = (\textbf{x}, \textbf{w})$.

---

### Official Review · AnonReviewer3 · 2019-10-22
**Official Blind Review #3**

**Rating:** 3

**Review:**

This paper proposed differentiable Bayesian neural networks (DBNN) to speed up inference in an online setting. The idea is to combine online vector quantization with Bayesian neural networks (BNN), so that only incremental computation is needed for a new data point.

It seems the idea of online codevector histogram, or online vector histogram, is not new, which is not surprising since the original codevector histogram work is from decades ago (1982). A quick search shows several related work in this domain, e.g., ‘An Online Incremental Learning Vector Quantization’. It would be better if the authors could clarify the differences if they want to claim the contribution.

One of my major concerns is the fairness in terms of comparison to BNN approaches. For example, in Table 2, BNN (shown as MU) is significantly slower than DU/DBNN and DNN. The authors mentioned that this is because MU predicts results for 10 batches of size 3, and therefore 30 times slower. Since DU and DBNN also uses MC-dropout, why is this not an issue for DU/DBNN. Such large overhead is also inconsistent with the description in Section 5.3, saying that the ‘overhead of sampling weights is negligible’. Could the authors elaborate on this?

A related problem for clarification: it is mentioned before Section 5.1 that 30 samples are drawn from the BNN’s posterior. Do you mean 30 feedforward passes of MC-dropout, as done in the MC-dropout paper? Also, the authors should have made it clear that they are using MC-dropout as a BNN.

In the introduction, the authors motivate the proposed DBNN by saying that BNN needs dozens of samples from weight distributions and therefore is rather inefficient. However, there are a lot of modern BNN that are both sampling-free and differentiable. For example, natural-parameter networks (NPN) parameterize both the weight and neuron distributions using natural parameters. Even the earlier work, probabilistic BP (PBP), as cited in the current paper, also counts as sampling-free. The claim of ‘being differentiable’ without acknowledging prior work is rather misleading.

The point above is also related to the MU baseline in Table 2. The issue of needing 30 passes can be readily resolved if modern BNN such as NPN (or PBP), which takes only one pass, is used.

The organization could be improved to make the paper more readable. It would be better if the problem setting of online inference is introduced at the beginning, followed by the overview of DBNN and then the OCH details. Otherwise, it is not clear what the focus of DBNN is, until Section 4.

Minor:

It might be better to denote the weight as ‘w’ rather than ‘x’, to avoid confusion.


**Experience Assessment:**

I have published in this field for several years.

**Review Assessment: Checking Correctness Of Derivations And Theory:**

I assessed the sensibility of the derivations and theory.

**Review Assessment: Checking Correctness Of Experiments:**

I carefully checked the experiments.

**Review Assessment: Thoroughness In Paper Reading:**

I read the paper thoroughly.

---

> ### Author Response · Authors · 2019-11-11
> **We appreciate your insightful comments (1/2)**
>
> Thank you for insightful and constructive comments.
>
> As you pointed out, online codevector is not new. The online codevector histogram (OCH) is a combination of the vector quantization histogram [Kotani, 2002] and stochastic variant of the algorithm which adds and deletes prototypical vectors, e.g. ILVQ [Xu, 2012] and Growing Neural Gas [Frezza-Buet, 2014],  depending on the data stream. Thus, we agree that the novelty of the OCH algorithm is incremental.
>
> In this paper, however, we want to emphasize how to use this rather than the algorithm of OCH itself. As pointed out in [Gal, 2016; Mohamed, 2019], three methods have been widely used to calculate the gradient of MC estimation: re-parametrization trick, score function, and measure-valued gradient estimation. However, as described in Section 2.2 using the re-parametrization trick as an example, these methods are computationally inefficient for DBNN. In short, OCH does not change the positions of codevectors. Instead, it only adds and removes codevectors and changes its weights. Thus, OCH calculates the difference of MC estimation by calculating the difference of the weights of the samples; this method is computationally efficient.
>
> References:
> - Kotani, Koji, Chen Qiu, and Tadahiro Ohmi. "Face recognition using vector quantization histogram method." Proceedings. International Conference on Image Processing. Vol. 2. IEEE, 2002.
> - Xu, Ye, Furao Shen, and Jinxi Zhao. "An incremental learning vector quantization algorithm for pattern classification." Neural Computing and Applications 21.6 (2012): 1205-1215.
> - Frezza-Buet, Hervé. "Online computing of non-stationary distributions velocity fields by an accuracy controlled growing neural gas." Neural Networks 60 (2014): 203-221.
> - Gal, Yarin. Uncertainty in deep learning. Diss. PhD thesis, University of Cambridge, 2016.
> - Mohamed, Shakir, et al. "Monte Carlo Gradient Estimation in Machine Learning." arXiv preprint arXiv:1906.10652 (2019).
>
>
> As you pointed out, sampling-free BNNs are one of the major research areas of BNNs. In this revision, we added the sampling-free BNN in related work (Section 2).
>
> Several sampling-free BNNs, e.g. PBP [Hernández-Lobato, 2015], NPN [Wang et al., 2016; Hwang et al., 2019], and DVI [Wu, 2018], have been proposed recently. The sampling-free BNNs predict results with only one forward pass. However, these methods have limitations. First, sampling-free BNNs necessitate specific types of weight-distribution with parameters such as Gaussian distribution or exponential family; in some cases, they assumes that the weights are factorized. When a posterior has multi-modality or skew, they can give inaccurate approximations. If NN is deep, there is no guarantee that the accumulative error is bounded. Second, in many cases, sampling-free BNN uses a method other than variational inference (VI) to obtain a posterior because evidence lower bound (ELBO) is not amenable. PBP uses expectation propagation (EP), which generally gives a blurry posterior compare to VI. When weights are multimodal, VI predicts reasonable results (with mode collapse), while the EP averages all modes, which does not give the correct results in most cases.  For these reasons, with the exception of some cases such as [Hwang et al., 2019], sampling-free BNNs have been applied little to deep and wide NNs in real-world situation, while sampling-based BNNs have been extensively tested in various fields. Therefore,  we mainly consider sampling-based BNN for comparison in this paper.
>
> To the best of our knowledge, DBNN is the first sampling-based BNN inference that requires one (or less) feedforward pass. DBNN including sampling-based BNN has the following advantages over sampling-free BNN. First, DBNN does not depend on the depth of NN. As shown in this manuscript, DBNN containing U-Net using 27 convolution blocks shows predictive performance comparable with BNN. Second, DBNN only depends on the samples from the posterior, not the shape of the posterior. This means that, as pointed out in main contributions, DBNNs cover arbitrary posteriors — even nonparametric or even discrete representations — unlike sampling-free BNNs. In other words, DBNN does not depend on training techniques.
>
> References:
> - Hernández-Lobato, José Miguel, and Ryan Adams. "Probabilistic backpropagation for scalable learning of bayesian neural networks." International Conference on Machine Learning. 2015.
> - Wang, Hao, S. H. I. Xingjian, and Dit-Yan Yeung. "Natural-parameter networks: A class of probabilistic neural networks." Advances in Neural Information Processing Systems. 2016.
> - Hwang, Seong Jae et al. “Sampling-free Uncertainty Estimation in Gated Recurrent Units with Applications to Normative Modeling in Neuroimaging.” UAI (2019).
> - Wu, Anqi, et al. "Deterministic variational inference for robust bayesian neural networks." (2018).

---

> ### Author Response · Authors · 2019-11-11
> **We appreciate your insightful comments (2/2)**
>
> As you suggested, to address the fairness of the evaluation, we added more data in the Appendix E.
>
> In this paper, all experiments use sampling-based BNNs, requiring multiple forward passes. Section 5.2 uses a fully-Bayesian NN, i.e., all weights are random variables, and Section 5.3 which is semantic segmentation experiment uses BNN with MC-dropout layers. (We mentioned only in Section 5.3 that the BNN contains the MC-dropout layers, but this can be confused with the description of the baseline. So, in this revision, we added the description to the baseline.)
>
> In the semantic segmentation experiment, "overhead of sampling weights is negligible" means "overhead of dropout layer is negligible in single forward pass" (we updated it in this revision). Thus, even though the BNN has additional dropout layers on the DNN, the execution time of BNN with 30 forward pass is just 30 times that of the DNN.
>
> DBNN also includes MC-dropout layer like BNN. However, while BNN executes repetitive NN on one data, DBNN executes only one forward NN pass. DBNN uses OCH to accumulate recent predictions, and it uses the distribution of these memorized predictions as well as the most recent prediction to predict the result and the uncertainty. In conclusion, past predictions are used to reinforce predictive uncertainty.
>
> The following questions may arise: “Can BNN with a small number of forward pass get uncertainty equivalent to DBNN?” Table 4 added in this revision shows computational and predictive performance for the number of forward pass of BNN (shown as MU). According to this table, to achieve the same uncertainty measures (Acc-90, Unc-90, IoU-90, NLL) as DBNN, MU requires 10 forward passes. In particular, Unc-90 — probability that the pixel is not 90% confident pixel when prediction is incorrect, i.e., $p(\textbf{unconfident} | \textbf{inaccurate})$ — of DBNN has always outperformed that of MU. For the improvement rate from Acc and IoU to Acc-90 and IoU-90, the uncertainty of DBNN is the same as that of MU with 30 forward passes.
>
>
>
>
> Also, according to your suggestions, we added some detailed descriptions to introduction and changed the notation as follows to avoid confusion: data $\textbf{x}_0$ → $\textbf{x}$, weight $\textbf{x}_1$ → $\textbf{w}$, and $\textbf{x} = (\textbf{x}_0, \textbf{x}_1)$ → $\textbf{z} = (\textbf{x}, \textbf{w})$.

---

### Official Review · AnonReviewer2 · 2019-10-23
**Official Blind Review #2**

**Rating:** 8

**Review:**

The paper proposes a differentiable Bayesian neural network. Traditional BNN can model the model uncertainty and data uncertainty via adding a prior to the weight and assuming a Gaussian likelihood for the output y. However, it's slow in practice since evaluating the loss function of BNN involves multiple runs over the entire network. Also when the input data is non-stationary, the output function can not be differentiated with respect to the input data. The paper proposes to use an online code vector histogram (OCH) method attached to the input and output of a classical DNN. Input OCH captures the distribution of both input data and network weights, and the output OCH captures the distribution of the predictions. Since these OCHs are differentiable, the proposed DBNN model can be used for streaming input data with time-variant distributions.

I think the idea is interesting and novel. It explores a different way of modeling distributions with DNN. Instead of adding priors, DBNN relies on histograms, which is usually used to describe distributions for discrete observed input data. So the paper is well-motivated.

1. The paper needs more literature review in the area of data streaming. Papers, such as [1], have proposed to use a vector quantization process that can be applied online to a stream of inputs. This paper introduces the vector quantization but doesn't mention the use of it in streaming data in related work, which kind of blurs the contribution a bit. Moreover, it would be helpful for readers to learn about useful techniques for streaming data from this paper.

[1] Hervé Frezza-Buet. Online computing of non-stationary distributions velocity fields by an accuracy controlled growing neural gas

2. I think the paper might need a bit more explanation about codevector, since it's not a very well-acknowledged concept in this field. The main issue for me to understand it is how to get these codevectors. When DBNN deals with streaming data and starts from no input, is the set of codevector empty at the beginning? The input data points are accumulated as codevectors? I hope the authors could clarify this process a bit more.

3. Given the insufficient understanding of codevector, figure 2 is a bit hard to read. 1) (a)-(d) are figures for x0 at t=0, which is not time-variant. 2) what are these codevectors picked. 3) It seems that the codevectors are out of the regime of the distribution of y. But according to algorithm 2, y_*<-T(c_*), would that be a problem? I think (a)-(d) are informative but not straightforward to read. The authors need to put more text to explain these figures, since this simulated example can help readers to understand what is codevector and how it helps for uncertainty estimation.

Overall I think the paper is well-written. The idea is novel and practical in the scenario of DNN. I would vote for accept.



**Experience Assessment:**

I have published one or two papers in this area.

**Review Assessment: Checking Correctness Of Derivations And Theory:**

I assessed the sensibility of the derivations and theory.

**Review Assessment: Checking Correctness Of Experiments:**

I assessed the sensibility of the experiments.

**Review Assessment: Thoroughness In Paper Reading:**

I read the paper at least twice and used my best judgement in assessing the paper.

---

> ### Author Response · Authors · 2019-11-11
> **Thank you for thoughtful and comments.**
>
> Thank you for your comments and suggestions.
>
> 1. According to you suggestions, we included  [Frezza-Buet, 2014], in Section 3, which is closely related to our online codevector histogram and cited other related work.
>
> 2. We agree to your point that our explanation about codevector was not enough. We added the following to this revision: Initially, OCH is empty and has no codevector. OCH adds input data as a codevector with high probability if the number of codevectors is small. After a period of time, OCH contains codevectors, which are the recent data from data stream.
>
> 3. Thank you for the constructive feedback.
> In this revision, we added more explanation for Figure 2 in Section 5.1. (We changed the notation as follows to avoid confusion: data $\textbf{x}_0$ → $\textbf{x}$, weight $\textbf{x}_1$ → $\textbf{w}$, and $\textbf{x} = (\textbf{x}_0, \textbf{x}_1)$ → $\textbf{z} = (\textbf{x}, \textbf{w})$)
>
> Simple linear regression experiment in Section 5.1 shows the difference between DNN, MU, DU and DBNN. The top of (a)-(d) in Figure 2 are approximated distributions of data and weight samples $p(x, w)$, and bottom are approximated distributions of output samples (with data) $p(x, y)$. To predict result, DNN uses point-estimated weight and data $x = 0$, which is the data at $t = 0$. MU uses weight distribution, instead of point-estimated weight, and data $x = 0$. DU uses point-estimated weight, which is the same as DNN. However, since DU contains OCHs, it uses not only the most recent data ($x = 0$ at $t = 0$) but also past data ($x < 0$ when $t < 0$). As a result, DU predicts the distribution of y. DBNN predicts y using both weight distribution and data from the past to now.
> 1) The simple linear regression experiment in this submission is slightly different from the typical regression: x is time-variant and is given by $p(x | t) = \delta(x - v t)$, so $p(x, w) = p(x)p(w)$ is time-variant ($p(w)$ is time-invariant). At the same time, y is also time-variant. The input probability $p(x, w)$ and output probability $p(y)$, i.e., marginalized $p(x, y)$ over x, are time-variant and (a)-(d) are snapshots at $t = 0$.
> 2) The data vector x is from the input data stream. NN parameter w is sampled from the posterior. OCH_Z, which represents the distribution of the input vector z, picks the x and the w with probability.
> 3) In this experiment, as you mentioned, the distribution of y obtained by DU and DBNN is biased. This is because OCH_Z and OCH_Y do not forget past codevectors fast enough because the data stream changes so fast; we set an extremely high data stream change speed to show the difference between each method.

---

### Decision · Program_Chairs · 2019-12-19

**Decision:**

Reject

**Comment:**

The main contribution is a Bayesian neural net algorithm which saves computation at test time using a vector quantization approximation. The reviewers are on the fence about the paper. I find the exposition somewhat hard to follow. In terms of evaluation, they demonstrate similar performance to various BNN architectures which require Monte Carlo sampling. But there have been lots of BNN algorithms that don't require sampling (e.g. PBP, Bayesian dark knowledge, MacKay's delta approximation), so it seems important to compare to these. I think there may be promising ideas here, but the paper needs a bit more work before it is to be published at a venue such as ICLR.